# Subglacial lake activity beneath the ablation zone of the Greenland Ice Sheet

Yubin Fan[1,2,3], Chang-Qing Ke[1,2,3*], Xiaoyi Shen[1,2,3], Yao Xiao[1,2,3], Stephen J. Livingstone[4], Andrew J. Sole[4]

[1]Jiangsu Provincial Key Laboratory of Geographic Information Science and Technology, Key Laboratory for Land Satellite Remote Sensing Applications of Ministry of Natural Resources, School of Geography and Ocean Science, Nanjing University, Nanjing, 210023 China.
[2]Collaborative Innovation Center of Novel Software Technology and Industrialization, Nanjing, 210023 China.
[3]Collaborative Innovation Center of South China Sea Studies, Nanjing, 210023 China.
[4] Geography Department, University of Sheffield, Sheffield, UK, S10 2TN.

*Correspondence to*: Chang-Qing Ke (kecq@nju.edu.cn)

**Abstract.** Hydrologically active subglacial lakes can drain large volumes of water and sediment along subglacial pathways, affecting the motion and mass balance of ice masses, and impacting downstream sediment dynamics. Only eight active lakes have been reported beneath the Greenland Ice Sheet (GrIS) to date, and thus the understanding of their spatial distribution and dynamic processes is still lacking. Here, using ICESat-2 ATL11 data, we identify 18 active subglacial lakes, 16 of which have not been previously reported. Multi-temporal ArcticDEM strip maps were used to extend the timeseries to verify lakes and determine their drainage history. The identification of active subglacial lakes beneath the GrIS is complicated by the occurrence of supraglacial lakes, which also fill and drain, and are hypothesized to be almost co-located. We therefore used the temporal pattern of ice-surface elevation change to discriminate subglacial lakes, and utilized the ability of ICESat-2 to penetrate through surface water to correct the elevation provided by the ATL11 data. A significant localized elevation anomaly (-16.03−10.30 m/yr) was measured in all detected subglacial lakes after correction, revealing that 6 subglacial lakes are twinned with supraglacial lakes. The active subglacial lakes have large upstream hydrological catchments and are located near or below the equilibrium line. Lakes have a median area of 1.20 km$^2$, and 12 lakes exhibited positive elevation-change rate during the ICESat-2 period. These observations illustrate the potential for combining ICESat-2 and the ArcticDEM to differentiate small subglacial lakes in the ablation zone and beneath supraglacial lakes.

## 1 Introduction

Subglacial lakes that fill and drain on annual to decadal timescales are termed hydrologically active subglacial lakes (henceforth 'active'). These lakes transiently store and then release water downstream, lubricating the ice-bed interface and affecting ice sheet mass balance by changing the ice discharge speed (Siegfried & Fricker, 2018; Malczyk et al., 2020). Some active subglacial lakes are hydraulically connected to other lakes, and water exchange between lakes can impact hydraulic gradients and subglacial water flow (Smith et al., 2017). Lake drainage not only exchanges water between lakes, but also transfers sediment and nutrients downstream, feeding microbial communities (Vick-Majors et al., 2020). Water crossing the

grounding line can also reduce the stability of ice shelves (Li et al., 2021). Therefore, knowledge of the distribution and water budget of active subglacial lakes is vital for understanding the stability of ice sheets.


Subglacial lakes can be identified from various remote sensing techniques. Gravity and seismic data using acoustic impedance or amplitude-versus-angle analysis can determine their bathymetries and characterize their geological properties (Studinger et al., 2004; Yan et al., 2022). Additionally, some stable subglacial lakes produce a flat ice-bed interface with high reflectance in radargrams and can therefore be recognized from radar echo sounding (RES) (Wright et al., 2012; Palmer et al., 2013; Bowling

et al., 2019; Bessette et al., 2021; Maguire et al., 2021). Water that moves in and out of subglacial lakes can lead to localized ice-sheet surface deformation, enabling the lakes' corresponding volume changes to be estimated through localized elevation anomalies detected from satellite radar (Siegfried & Fricker, 2018), laser altimeters (Smith et al., 2009; Siegfried & Fricker, 2021), and multi-temporal optical (Palmer et al., 2015) and radar interferometry (Gray et al., 2005).

More than 675 subglacial lakes have been detected underneath the Antarctic Ice Sheet (Livingstone et al., 2022), including more than 130 active lakes (Smith et al., 2009; 2017). Conversely, only 8 active and 57 stable subglacial lakes have been identified underneath the Greenland Ice Sheet (GrIS) (Livingstone et al., 2022; Sørensen et al., 2023), although hydrologic potential calculations indicate that subglacial lakes could account for approximately 1.2% of the GrIS area (Livingstone et al., 2013). Constrained by steeper ice surface slopes and thus stronger hydraulic gradients, lakes underneath the GrIS tend to be

smaller (Bowling et al., 2019), making it difficult for satellite altimeters (e.g., ICESat and CryoSat-2) to study subglacial lake activity in detail due to their coarse spatial or temporal resolutions. The few active lakes underneath the GrIS that have been observed, were identified from multi-temporal Digital Elevation Models (DEMs) (Palmer et al., 2015; Howat et al., 2015; Bowling et al., 2019; Livingstone et al., 2019; Sørensen et al., 2023).

ICESat-2 has an improved footprint size (approximately 11 m with 0.7 m along-track spacing) (Magruder et al., 2020) and spatial coverage (±88° latitudes) compared to previous satellite altimeters, providing an essential dataset for enabling active subglacial lake detection across the GrIS. Furthermore, its 91-day revisit cycle has the ability to reveal how the basal water system operates on sub-annual timescales. This study aims to detect active GrIS subglacial lakes by measuring ice-surface elevation anomalies observed from ICESat-2 between March 2019 and December 2020. Subglacial lakes were verified and

their boundaries identified using the ArcticDEM (Porter et al. 2018). Spatial patterns of elevation and volume changes over the ICESat-2 period (2019-2020) were generated, and the elevation time-series over the combined ArcticDEM and ICESat-2 periods (2009-2020) were used to determine the temporal patterns of lake activity.

## 2 Data

### 2.1 ICESat-2 data

The ATL11 product 'Slope-Corrected Land Ice Height Time Series' (Smith et al., 2021) is derived by correcting offsets between the reference ground track (RGT) and the location of ATL06 land ice measurements, and provides land ice surface elevations with a 91-day cycle in polar regions (poleward of 60° N and 60° S), accompanied by geolocation information and the corresponding quality assessment. The three beam pairs of ICESat-2 follow a reference pair track (RPT) parallel to the RGT, with the reference points of the ATL11 product spaced along each RPT. The ATL11 product is posted at a spatial resolution of 60 m, with the spacing of tracks within each RPT ranging from approximately 5.4 km (high latitude) to 7.4 km (low latitude). More information on the ATL11 data and its processing algorithm can be found in Smith et al. (2021).

The ICESat-2 ATL11 v3 product contains ice surface elevations with respect to the WGS84 ellipsoid from March 2019 to December 2020 (i.e., cycles 3-9). In total, the elevation measurements of all 511 RGTs (1533 RPTs) that intersect Greenland were used to detect active subglacial lakes and explore their elevation and volume changes from 2019 to 2020. We collated $2.91 \times 10^7$ reference points over the Greenland Ice Sheet, and only data with cycles marked as good quality (quality_summary=0) were used.

The ATL03 product 'Global Geolocated Photon Data' contains latitude, longitude, elevation, and time for all photons collected by the Advanced Topographic Laser Altimeter System (ATLAS) on board ICESat-2. The ATL06 (i.e. land ice height) and ATL11 products only capture the elevations of the top photons (and thus identify the ice or water surface only), but the ATL03 data contain the full stream of returned photons (Neumann et al., 2019), which were used to identify surface meltwater depths and correct the ice surface elevation measurements for its presence during the melt season.

### 2.2 Verification data

The ArcticDEM is a high-resolution, high-quality digital surface model (DSM) of the Arctic based on optical stereo imagery from GeoEye-1 and WorldView-1/2/3 (Porter et al., 2018), and with an internal accuracy of 0.2 m (Noh & Howat, 2015). The 2-m resolution strip DSM files provided time-stamped elevation measurements from August 2009 to March 2017. The temporal resolution of these time-stamped DSM segments is variable due to the influence of clouds and shadows. Nevertheless, the dataset enables the detection of localized elevation-change anomalies, and was used for lake cross- verification and boundary estimation. Published Greenland subglacial lake locations (Livingstone et al., 2022), comprising active lakes detected from multi-temporal ArcticDEM elevation change and stable lakes detected from RES data, were also used for verification.

## 3 Methods

### 3.1 Identification of active subglacial lakes

Subglacial lakes were detected from localized ice-surface elevation anomalies measured by ICESat-2. The elevation-change
rate of individual reference points was obtained through a linear fit by using the timestamp and elevation value of valid
elevation measurements (e.g., Figure 1a). We then generated a Greenland-wide elevation change trend map by gridding these
point trend data at a resolution of 500 m, which covers approximately 80% of the GrIS. The change map was used to create
masks for candidate regions. Previous studies in Antarctica used a threshold of ± 0.5 m/yr to select regions with a significant
localized elevation change (Fricker et al., 2007; Smith et al., 2017; Malczyk et al., 2020), but knowledge of such a threshold
applicable to Greenland subglacial lakes is lacking. We adopted a more conservative threshold of ± 0.2 m/yr to identify
potential subglacial lakes that could then be verified using the ArcticDEM dataset and through manual examination of ice-
surface elevation patterns.

The relative elevation-change anomaly associated with a subglacial lake should have a characteristic spatial pattern comprising
an obvious elevation anomaly at the lake center which reduces to zero (within uncertainty) outside the lake. Candidate regions
where such elevation anomalies can be explained by other factors, including displacement of the ICESat-2 footprints, dynamic
topography, and cloud cover, etc., were discarded (Smith et al., 2009). Displacement of the ICESat-2 tracks was corrected by
the ATL11 product itself, and the slope generated by the mosaicked 100-m ArcticDEM product, which was used as a
topographic reference. The elevation profiles were used to determine lake location by visual interpretation (Willis et al., 2015)
(e.g., Figures 1b, c), and the profiles that exhibited gradual elevation change with time were retained.

### 3.2 ArcticDEM verification, lake boundary determination and lake activity recognition

Time-series of time-stamped ArcticDEM data were used to cross-verify subglacial lake locations. We only used DSM strips
where correction vectors obtained by the co-registration between filtered ICESat altimetry data were provided within the
metadata. Areas of known subglacial lakes in Greenland range from 0.18 to 8.4 km$^2$, with a maximum length of 1.6 km
(Livingstone et al., 2022). Therefore, a 5 km radius circular buffer was established around the point at the center of the potential
lake determined from the ICESat-2 data, which was taken as the maximum possible extent of the subglacial lake. To provide
spatially continuous images and improve computational efficiency, we derived the median value of the DSMs every 100 days
to obtain elevation maps. Then, we calculated the elevation difference between each temporally adjacent elevation map, which
was used to determine whether there was an elevation anomaly (e.g., Figure 1a). Elevation anomalies identified in both the
ICESat-2 and ArcticDEM data were confirmed as potential subglacial lakes (henceforth 'confirmed lakes'). We acknowledge
that the time differences between ICESat-2 and ArcticDEM data might affect the percentage of confirmed lakes because some
lakes did not exhibit complete drainage or filling activity. However, it allowed us to extend the temporal coverage of the data
by 8 years, giving a more comprehensive picture of the patterns of elevation changes, which was critical for discriminating
subglacial lakes from other processes (e.g., supraglacial lake filling and draining). The large spacing of ICESat-2 tracks (5-7

km, exceeding the lake size) make it difficult to extract the subglacial lake boundary by generating an elevation-change surface

through interpolation of the ICESat-2 data. Therefore, lake boundaries were manually delineated from the ArcticDEM elevation-change anomaly maps. We still retained subglacial lakes that were not identified from the ArcticDEM (henceforth 'unconfirmed lakes'), to analyze the spatial pattern and elevation-change rate, but eliminated them from our analysis of volume change.

Time-series of elevation change were used to determine subglacial lake fill-drain patterns. We sampled the ArcticDEM DSMs

at the locations of the ATL11 measurements to construct a self-consistent time series, and calculated the relative elevation anomaly by subtracting the averaged ATL11 elevations within the lake outline from the buffer around it to remove the influence of systematic vertical and horizontal offsets between ArcticDEM tiles (Livingstone et al., 2019). A Hampel filter is a type of outlier detection filter commonly used in data analysis, and it works by identifying data points that are significantly different from their neighboring data points and replacing them with a more representative value based on the surrounding data (Hampel,

1974). Therefore, it was used to remove the outliers in the time-series of the combined ArcticDEM and ICESat-2 periods (2009-2020), which was used to determine the temporal patterns of lake activity (e.g., Figure 1d). For calculating the relative elevation anomaly, we used the relative errors of the data as a measure of uncertainty. The relative error of the ArcticDEM is 0.2 m (Noh & Howat, 2015), and 0.04 m for ICESat-2 footprints (Brunt et al., 2021).

### 3.3 Impact of supraglacial lakes on the detection of subglacial lakes

Numerous supraglacial lakes seasonally form in much of the ablation zone of the GrIS, and then either freeze or drain over the ice surface or to the bed (Selmes et al., 2011). The filling and drainage of these lakes produces ice-surface elevation anomalies in the ATL06 product (and therefore the ATL11 product) that could be mis-classified as subglacial lake activity. This is particularly challenging because supraglacial and subglacial lakes are hypothesized to exist in tandem (Sergienko, 2013). Moreover, if a subglacial lake located beneath the ablation zone drains, the ice-surface depression created would provide a

natural basin for water to pond (Willis et al., 2015).

To discriminate between surface and subglacial lakes we first evaluated the temporal pattern of the ice-surface elevation changes. Supraglacial lakes often drain rapidly to the bed in the summer via moulins (MacFerrin et al., 2019), and are therefore characterized by a seasonal fill-drain pattern, whereas subglacial lakes tend to fill over multiple years. Fair et al. (2020) demonstrated that the ICESat-2 ATL03 photon data produces a double reflection of the water surface and ice surface beneath.

ICESat-2 can therefore be used to monitor ice surface elevation changes under surface lakes, and it offers the ability to monitor ice-surface elevation changes caused by subglacial lake filling and draining (or other processes) independent of the confounding influence of supraglacial lakes above. For each potential subglacial lake, we identified whether there was a double reflection in the ATL03 profile, using Landsat-8 images around the acquisition time of ICESat-2 to manually check for the presence of surface water. Where a double reflection was identified, we applied the Watta algorithm (Datta et al., 2021) to

discriminate the lake bottom (ice surface) elevation. This approach does not work when surface lakes form ice lids (Figure 2).

However, only one lake was influenced by an ice lid and so this had a limited impact on lake identification. Watta-derived depths showed a high correlation with the LandSat-8/ Sentinel-2 -based and manual-picked depths (Fricker et al., 2020), and the depth uncertainty is small compared with the corresponding elevation change. The bottom elevation was taken as the corrected ATL11 elevation and used to recalculate the elevation-change rate for subglacial lake footprints within each supraglacial lake. This correction was applied to 6 subglacial lakes, and in all cases a significant localized elevation anomaly was still measured (Figure S1d).

### 3.4 Lake confidence level classification

We classified potential subglacial lakes into three confidence levels (e.g., Figure 3). Low confidence lakes exhibited no clear pattern of multi-year elevation change with time, might be associated with flat surfaces and annual elevation cycles that could be the expression of supraglacial lakes, had a limited number of data points, and exhibited abrupt elevation changes. High confidence lakes were identified by a gradual elevation change over time, had a clear double reflector or no evidence of surface water, and an elevation change pattern typical of subglacial lakes (e.g., multi-year pattern of filling and then rapid drainage). Medium confidence lakes had an elevation change pattern typical of subglacial lakes, but a less clear signal, for example a smaller ice-surface elevation change, fewer data points or some flat surfaces. We discounted the low confidence potential lakes as likely to be caused by other processes (e.g., filling and draining of supraglacial lakes).

### 3.5 Estimation of lake elevation and volume change

The elevation-change rate within the lake polygons is composed of ice-flux divergence, ice ablation and basal water motion (Smith et al., 2009), while the ice outside is only affected by ice-flux divergence and ablation. This 'background' elevation change needs to be subtracted to calculate the relative elevation-change caused by the subglacial lake. For each ICESat-2 overpass, we first calculated the median value of all ICESat-2 measurement points within the lake polygon, and then the median elevation of the area surrounding the lake (within the buffer-region) was subtracted to produce the elevation anomaly. To quantify the effect of buffer-region width on the calculated elevation-change rate, we tested three ring buffers which extended beyond the lake outline: buffer1, a fixed buffer of 2 km width; buffer2, a buffer with a width equal to the radius of a circle whose area is equal to the lake, and buffer3, with half the width of buffer2 (Table S1). The fixed 2 km buffer exhibited a large difference compared to the adaptive ones because most lakes are smaller than 1 km$^2$. The mean value of the absolute differences between the two calculated elevation-change rates using adaptive buffers was approximately 0.10 m, which only accounts for 2.4% of the averaged absolute elevation-change rate. Therefore, the effect of the buffer size on the elevation-change rate was neglected, and buffer2 was applied because it is a similar footprint size to the lake region. For the unconfirmed lakes, we used half of the ICESat-2 along-track distance where the elevation anomaly was detected as a buffer.

We calculated the corrected elevation change rate, dh$_c$ for each lake as shown in Equation 1:

$$\mathrm{dh_c} = \mathrm{dh}_{median,\text{inside}} - \mathrm{dh}_{median,\text{outside}} \tag{1}$$

where $dh_{median,inside}$ is the median elevation-change rate of ATL11 footprints within each lake's bounding polygon, and $dh_{median,outside}$ is defined as the median value of the elevation change rate for ATL11 footprints outside the bounding polygon but within the buffer zone.

The uncertainty of the elevation-change rate was calculated by the standard deviation of the elevation-change rates of all footprints inside and outside the lake polygon, defined in Equation 2.

$$dh_{c,uncertainty} = \sqrt{dh_{std,inside}^2 + dh_{std,outside}^2} \qquad (2)$$

The volume change rate was derived by integrating the elevation change rate and lake boundary for the confirmed lakes (Equation 3). To estimate the errors in our volume change estimates caused by boundary migration, we assumed an area uncertainty of one grid cell of the ArcticDEM differencing image (i.e., 30 m x 30 m) and calculated the volume change uncertainty as shown in Equation 4.

$$dV_{confirmed} = dh_c \times area \qquad (3)$$

$$dV_{confirmed,uncertainty} = \sqrt{(dh_{c,uncertainty} \times area)^2 + (dh_c \times area_{uncertainty})^2} \quad (4)$$

For the unconfirmed lakes we only calculated elevation change and its uncertainty because the boundaries could not be determined.

## 4. Results

### 4.1 Cross-verification of subglacial lake location

Using ICESat-2, we identified 6 high confidence and 12 medium confidence active lakes (18 in total). Elevation variation occurred in a total of 12 of these lakes during the ArcticDEM period, indicating long-term patterns that continued into the ICESat-2 period (Figure 4a). Two previously identified active subglacial lakes were also identified in this study, located at the Flade Isbink Ice Cap (Willis et al., 2015) and Inuppaat quuat (Howat et al., 2015; Palmer et al., 2015). Four of six other reported active lakes were sampled by ICESat-2, but no characteristic spatial pattern of subglacial lake filling and draining was found, indicating that these lakes may be transient features or have been in a relatively steady state during the ICESat-2 period. RES data collected during 1993-2016 were analysed by Bowling et al. (2019), revealing 57 stable lakes. Of the 57 stable lakes, 39 of them were sampled by the ICESat-2 ATL11 data (within a circular buffer with a radius half the lake length derived from Livingstone et al. (2022)), but no clear elevation anomalies were found. In addition, 3 of the 18 active lakes were sampled by RES data from 2017 to 2019, but no classic flat reflections were identified. This mismatch between RES- and altimeter-detected lakes has also been reported in Antarctica (Siegert et al., 2014).

## 4.2 Distribution of active subglacial lakes

In total, 927 ICESat-2 ATL11 reference points sampled active subglacial lakes identified over the entire GrIS, with 8 lakes covered several times, but by only one RPT. The well-sampled subglacial lakes covered by 3-4 RPTs are located in northernmost Greenland. We adopted informal names for identified subglacial lakes (Table S2) based on the associated Greenland basin name (Mouginot et al., 2019).

Active subglacial lakes are concentrated toward the ice margin and have large upstream subglacial hydrologic catchments
(Figure 4a). Three main clusters of active lakes were observed in southwestern, western and northern Greenland, corresponding to regions of significant negative surface mass balance (Khan et al., 2022) and where surface meltwater can access the bed due to limited firn and the occurrence of moulins and crevasses. This distribution is consistent with that predicted by Bowling et al. (2019), with hydrologically active lakes located near or below the Equilibrium Line Altitude (ELA). There is a general paucity of active lakes in the southeastern sector of Greenland where high accumulation rates and thick firn limit the amount
of surface-derived water that reaches the ice bed (Selmes et al., 2011), and inland sectors of Greenland, where the bed is thought to be largely frozen (MacGregor et al., 2022). In contrast, stable subglacial lakes tend to be located in northern and eastern regions above the ELA (Bowling et al., 2019). Active lakes are typically located near regions of fast ice flow (>50 m/yr) (Figure S2) and 14 of them are within marine-terminating catchments. This distribution is consistent with the spatial pattern found in Antarctica (Smith et al., 2009).

The active subglacial lakes identified in this study differ in size from those observed in Antarctica, reflecting the different topographic setting, and steeper ice-surface slopes and thus hydrologic gradients controlling the morphology of subglacial lakes (see also Bowling et al., 2019). Lake area ranges from 0.27 to 5.29 $km^2$, with a median area of 1.20 $km^2$, and five of the subglacial lakes have an area < 1 $km^2$ (Figure 4b). The areas of unconfirmed lakes were comparable to those of confirmed lakes based on analysis of their diameter along the ICESat-2 tracks.

## 4.3 Elevation change and water budget

Ice-surface elevation range is a proxy for subglacial lake depth. By combining the elevation time-series of the ArcticDEM and ICESat-2 data to give a maximum lake depth estimation for the 12 confirmed lakes, we show that 3 lakes have a depth greater than 50 m, including one known lake located beneath the Flade Isbink Ice Cap, and the estimated depth of this lake (approximately 79 m) is consistent with Liang et al. (2022) (Figure S3).

Generally, active subglacial lakes in Greenland exhibit higher elevation change rates (usually larger than 1 m/yr) than those in Antarctica. Positive temporal elevation trends were identified in 66% of the lakes detected during 2019-2020 (Figure 4c), indicating net water recharge. The absolute elevation-change rates ranged from 0.13 to 16.03 m/yr with a mean value of 4.13 m/yr. The uncertainty of the elevation-change rate generally depended on the number of footprints, the slope of the lake bed, and the acquisition time of different tracks, and ranged from 0.32 to 8.09 m/yr with a mean value of 3.14 m/yr. The elevation-

change uncertainty may be related to the empty depression on the ice surface gradually being refilled by ice flow and the moulins or crevasses that allow the water to escape from a closed depression in the ice surface.

Our ability to estimate subglacial lake volume changes depended on the location and size of the lake in relation to the ICESat-2 tracks that detected the elevation anomalies. Large lakes tended to have faster volume-change rates than small lakes (with a correlation coefficient of 0.66, $p < 0.01$), suggesting that they have a greater impact on the subglacial hydrological system.

Subglacial lake volume changes exhibited the same temporal pattern as the elevation changes, with most lakes displaying a positive volume change over the observation period of ICESat-2 (Figure 4d). The absolute volume change rates ranged from $8.1 \times 10^5$ to $1.66 \times 10^7$ m$^3$/yr with a mean value of $5.36 \times 10^6$ m$^3$/yr (Table S2). Volume change rate uncertainties ranged from $1.27 \times 10^5$ to $1.31 \times 10^7$ m$^3$/yr, with a mean value of $1.82 \times 10^6$ m$^3$/yr. Four hydrological basins exhibited a net volume gain, with the most significant gains located in basins 6.2 and 7.1 (Figure 4d).

## 5. Discussion

### 5.1 Dynamic processes of active subglacial lakes

Variable subglacial lake activity was detected by ICESat-2 during 2019-2020. Nine lakes exhibited only filling or draining throughout the study period. In contrast, 2 lakes experienced at least 3 filling or draining periods during 2019-2020 (Table S3). Recharge of these subglacial lakes is thought to be generated from geothermal heat flux, frictional heating from ice flow and surface meltwater inputs (Bowling et al., 2019). As all 18 active lakes are located near or below the equilibrium line in areas of high negative surface mass balance, we hypothesise that surface meltwater runoff that reaches the ice bed has a strong control on lake activity (see also Liang et al., 2022). A total of 60 lake filling events (positive volume change) occurred from 2009 to 2020, and the volume-change rates have a positive correlation (correlation coefficient of 0.40, $p < 0.01$) with the cumulative runoff estimates from the high-resolution Regional Atmospheric Climate Model (RACMO2.3p2) (Noël et al., 2018).

### 5.2 Lake activity: fill-drain patterns

Livingstone et al. (2022) classified subglacial lake activity into 5 temporal patterns based on the ratio of filling and draining durations. They found that 3 active subglacial lakes in Greenland exhibited quiescence at high stand. To further improve the understanding of dynamic hydrological processes underneath the GrIS, we used the combination of ArcticDEM and ICESat-2 to determine the fill–drain patterns of our identified active lakes over 11 years.

The temporal resolution of the ArcticDEM varies, making it difficult to discriminate clear fill-drain patterns for all lakes. However, two lakes remained filled for multiple years (i.e. quiescent at high stand) before rapidly draining, providing further support for an external threshold controlling the initiation of lake drainage in Greenland (Livingstone et al., 2022). We did not identify active subglacial lakes that exhibited similar rates of filling and draining or that remained drained or partially

drained for multiple years (i.e. quiescent at low stand) before filling and draining. Twelve of 14 drainage events happened between May and August (Table S3), with 2 lakes draining between December and February. The tendency for lakes to preferentially drain in summer also supports the idea that surface meltwater can influence or trigger drainage although there is a bias here with the acquisition data restricted to summer.

## 6. Conclusions

We used ICESat-2 altimetry to detect active subglacial lakes underneath the Greenland Ice Sheet and to discriminate their signal from supraglacial lake drainage patterns. Multi-temporal ArcticDEM strip maps were used to extend the timeseries to verify the lakes and quantify their drainage history. In total, we identified 16 new active lakes (increasing the number of active lakes in the Greenland inventory to 24). Lakes are concentrated below the ELA, and correspond with regions of significant negative surface mass balance. This spatial distribution indicates that the formation and dynamism of active subglacial lakes

in Greenland is related to the ability of surface-derived meltwater to access the ice bed (i.e., little snow/firn and lots of crevasses and/or moulins). Five of the subglacial lakes had an area $< 1$ km$^2$, and large lakes exhibited faster volume-change rates than small lakes, suggesting that they have a greater impact on the subglacial hydrological system. Finally, lake drainages typically occur in the summer melt season, and 2 lakes where clear fill-drain cycles were identified displayed long-term quiescence at high stand followed by drainage, suggesting surface melt might control the initiation of subglacial lake drainage in Greenland.


There is no doubt that our inventory is incomplete, likely missing lakes in the lower-latitude regions where the ICESat-2 track spacing is large, despite using the time-stamped ArcticDEM data to fill spatial gaps. Although discriminating between supraglacial and subglacial lakes remains a challenge in the detection of subglacial lakes, we demonstrate the utility of ICESat-2 for removing the influence of shallow supraglacial lakes. Our confidence in identifying subglacial lakes and their drainage

patterns will increase in the future as the temporal coverage is extended by ICESat-2 and other satellite data. Future work could use our inventory to determine the impact of subglacial lakes on the wider ice sheet system, including subglacial hydrology, ice dynamics and sediment and biogeochemical fluxes.

## Data availability

Thhe detail information of 18 active subglacial lakes can be downloaded from the National Tibetan Plateau/Third Pole

Environment Data Center, Institute of Tibetan Plateau Research, Chinese Academy of Sciences at https://data.tpdc.ac.cn/en/data/702726a9-c457-4bb5-bf15-0222564bff5d (DOI: 10.11888/Cryos.tpdc.273022).

**Author contribution**

Yubin Fan performed the identification of active lakes and wrote the manuscript; Chang-Qing Ke contributed to the conception of the study and supervised the work; Xiaoyi Shen contributed to the discussion and advised on the elevation change and water budget; Yao Xiao performed ArcticDEM validation on Google Earth Engine (GEE) platform. Stephen J. Livingstone and Andrew J. Sole revised the manuscript and advised on lake confidence level classification. All authors contributed to the discussion of the results and to the improvement of the manuscript.

**Competing interests**

The authors declare that they have no conflict of interest.

**Acknowledgements**

This work was supported by the Program for National Natural Science Foundation of China (grant No. 41830105 & 42011530120). ICESat-2 data (https://nsidc.org/data/ATL11/versions/3) were obtained from the National Snow and Ice Data Center. The ArcticDEMs were obtained and processed in the GEE platform (https://code.earthengine.google.com/). RACMO2.3p2 Greenland daily runoff data were kindly provided by Brice Noël and MAR data were kindly provided by Xavier Fettweis.

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

**Figures**

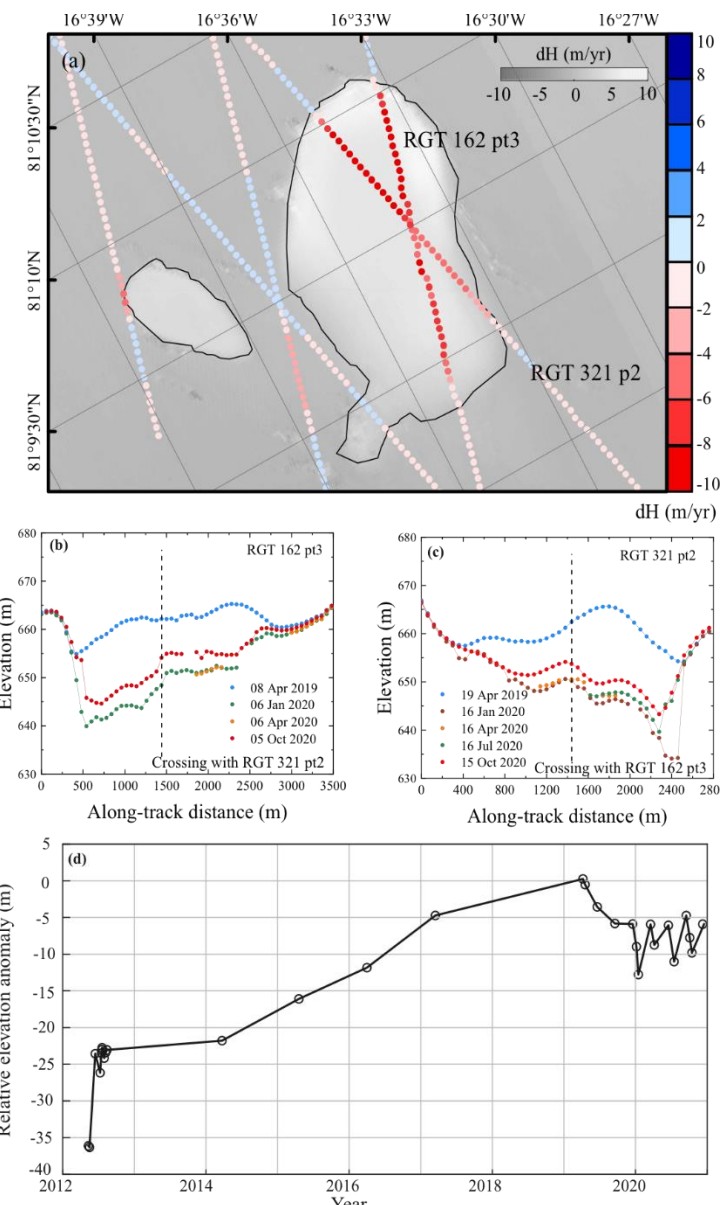


**Figure 1.** Active subglacial lake detection method, using Subglacial Lake ICE_CAPS_NE01 as an example. (a) The elevation change rate was derived from ICESat-2 and overlaid on the elevation difference maps between the two ArcticDEMs (20160923-20160621). The black polygons show the inferred lake boundaries derived from the ArcticDEM. The red-blue spots represent the elevation change rate derived from the linear fit of ICESat-2, while the grayscale colorbar represents the

elevation change rate derived from the ArcticDEM. Elevation anomaly profiles across the subglacial lake are given for RGT 162 pair track (pt)3 (b) and RGT 321 pt2 (c). The colors of the points correspond to the ICESat-2 observation times, and the vertical dashed lines show the location of the cross point. (d) Time-series of relative elevation anomaly based on the combined ArcticDEM and ICESat-2 tracks. Note the ~30 m of ice uplift over 6 years that is interpreted to be subglacial lake filling, followed by 10 m of subsidence over 1 year interpreted as slow subglacial lake drainage.


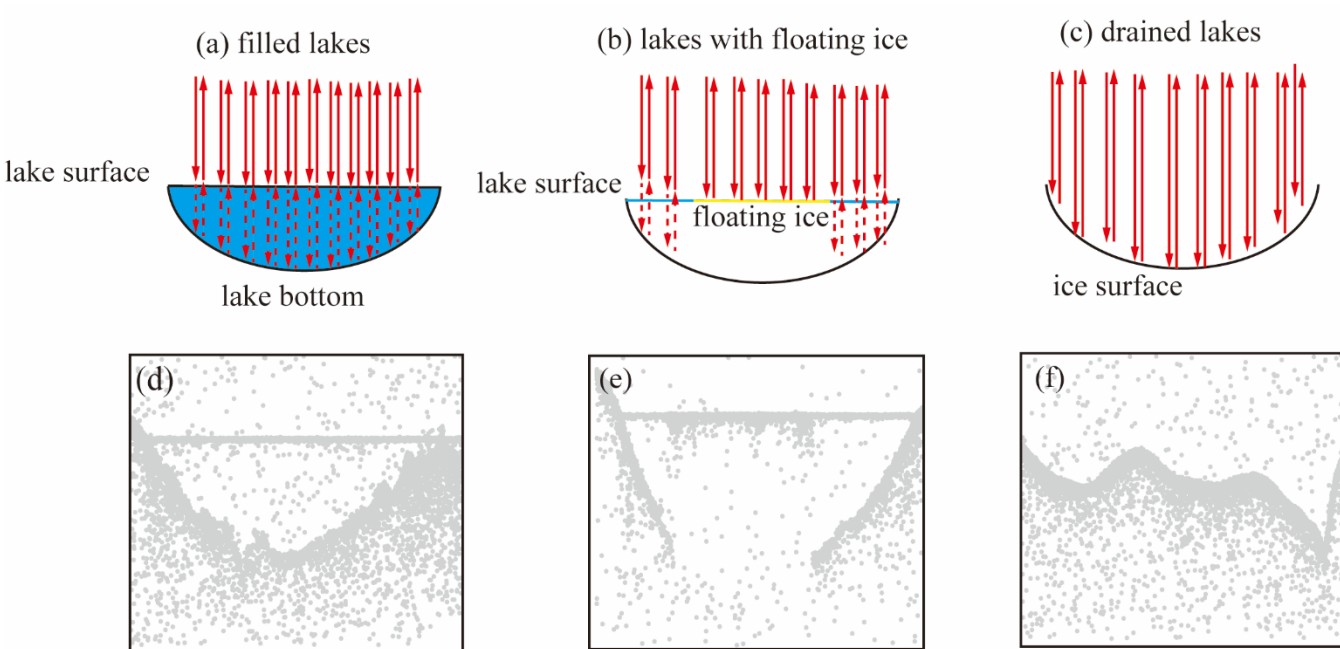

**Figure 2.** Example of how ICESat-2 can penetrate through the water column of supraglacial lakes to measure the lake bottom. The solid and dashed lines indicate strong and weak reflections, respectively. Figure (b) shows an example of a surface lake with floating ice. ICESat-2 can only penetrate the lake surface, but reflects directly off the floating ice. Figure (c) shows an example of a drained lake that ICESat-2 directly measures the ice surface. The second row (d-f) shows examples of the ICESat-

2 photon reflection for the corresponding schematic.

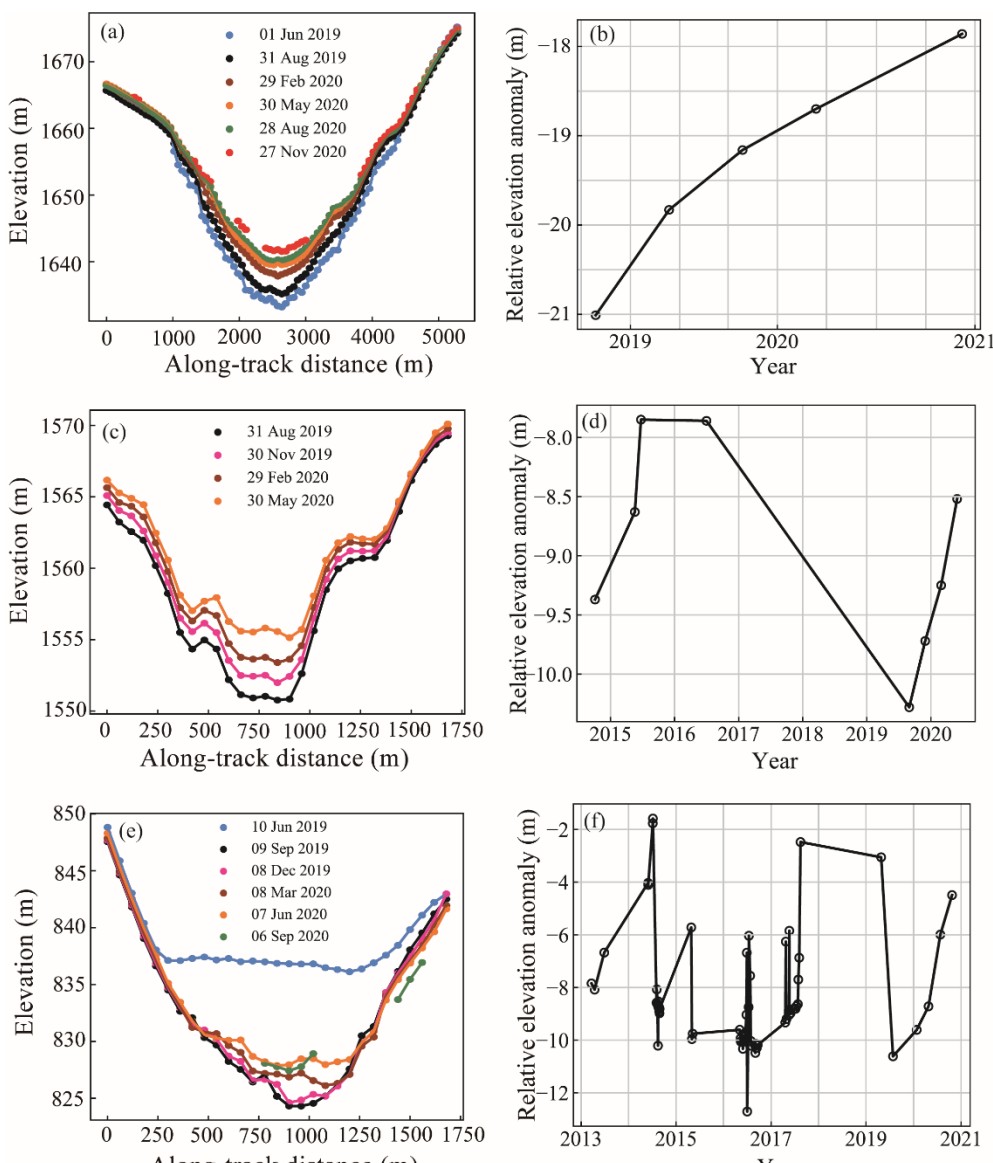

**Figure 3.** Examples of lake confidence level classification. The left column shows the elevation anomaly profiles across the subglacial lake derived from ICESat-2, and the right column shows the time-series elevation anomaly based on ArcticDEM and ICESat-2 tracks. The first row (Lake INUPPAAT_QUUAT02) is the high confidence level, which exhibits gradual ice-surface elevation change and an elevation change pattern typical of subglacial lakes. Second row (Lake KANGIATA_NUNAATA_SERMIA01) is a medium confidence level lake, which shows consistent elevation change, but with a less clear elevation-change pattern. The third row is a low confidence level lake, which contains a clear flat spot and a seasonal elevation change signal more typical of surface lakes and is therefore discounted from the inventory. More profiles can be found from Figure S4 to Figure S21.

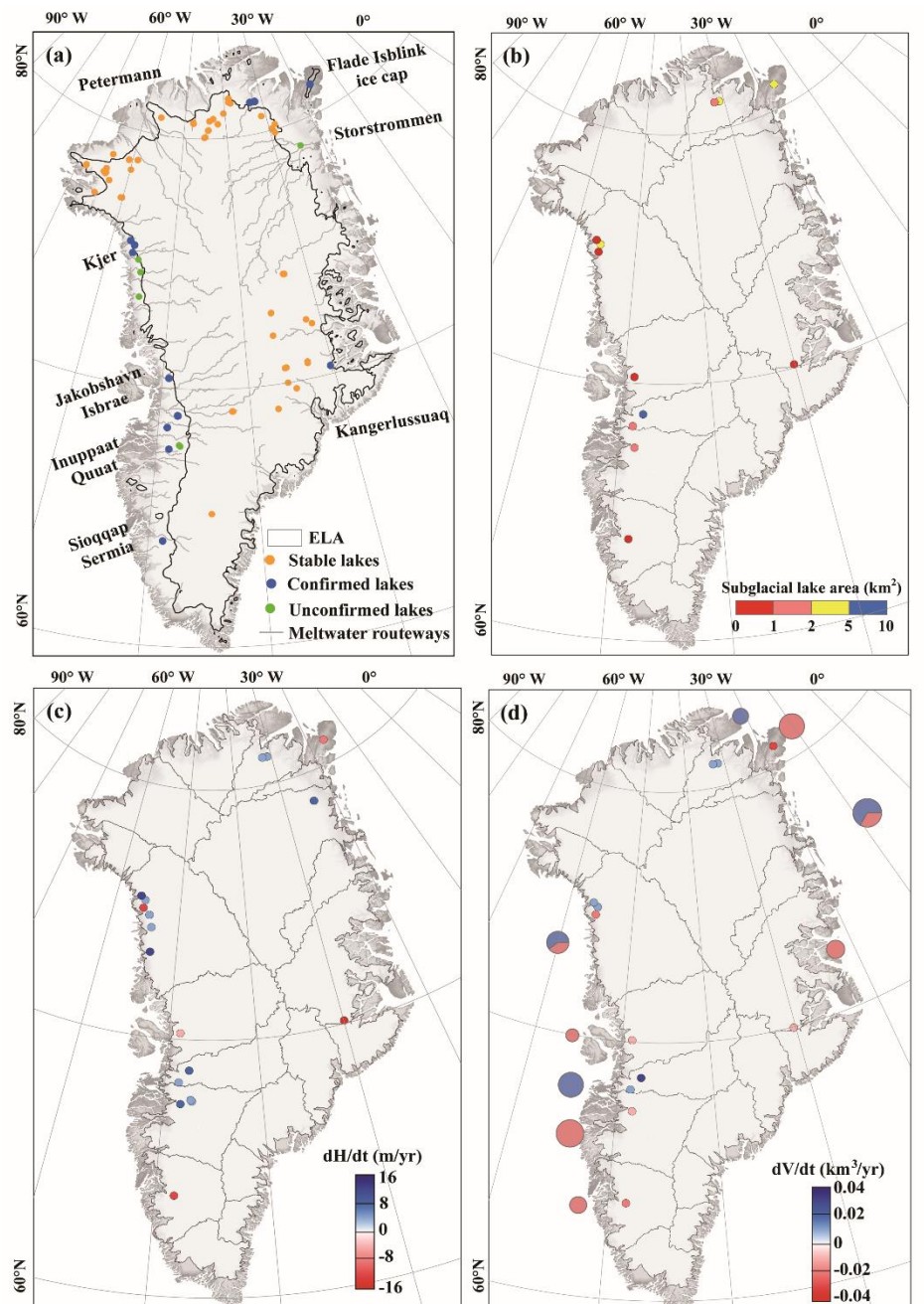

**Figure 4.** Maps of the (a) location, (b) area, (c) elevation change rate and (d) volume change rate during ICESat-2 period for the current active subglacial lakes under the Greenland Ice Sheet. The pie chart in panel (d) represents the overall volume-change rate of lakes in each basin (blue for positive, red for negative), with the pie size is proportional to the magnitude of the absolute rate. Meltwater pathways were derived from the hydraulic gradient (Livingstone et al., 2013). The Equilibrium Line Altitude (ELA) was derived from daily MARv3.12.1 data (Fettweis, et al., 2021). Stable lakes in (a) are from Livingstone et al. (2022).