# Peer review of "Subglacial lake activity beneath the ablation zone of the Greenland Ice Sheet"

_The Cryosphere, 2022_

## Author Comment (AC1)

Reviewer#1

Subglacial lakes are an important component of ice sheet hydrology, particularly in their influence on meltwater transfer between the ice sheet surface and bed. Although previous efforts have documented subglacial lakes over Antarctica, lakes over the Greenland Ice Sheet have received less attention. This manuscript aims to improve our understanding of Greenland subglacial lakes using ICESat-2 altimetry data. A combination of the ICESat-2 ATL11 product and the ArcticDEM is used to infer and validate the location of subglacial lakes. Interannual changes in ice sheet height from ATL11 are used to measure changes in lake area, height, and volume. The authors were able to identify 61 subglacial lakes over Greenland, many of which were not reported in previous literature.

Overall, this paper discusses an interesting and underreported topic in the cryosphere community, and its contents fall within the scope of The Cryosphere. The paper is generally well-written, and it is interesting that to see ATL11 applied to a novel application. Before publication, I do have a few concerns:

We thank the reviewer for the helpful feedback, we are appreciative of his or her help and time.

Main Comments

Previous methods for lake detection have used lower level ICESat-2 products, such as ATL03 or ATL06. Although it is interesting to see ATL11 used for this analysis, and ATL03 is used to identify supraglacial lakes, I would like to see some justification on why ATL11 was selected as the primary dataset over other ICESat-2 products. A reader less familiar with ICESat-2 may wish to know why ATL11 is preferable for this application, or why other ATLAS products may be less effective.

Response:

The ATL06 product is developed from the ATL03 product, and is corrected for

instrument bias (e.g. transmit pulse shape bias correction and first-photon bias correction). ATL11 is then based on the ATL06 product, which employs a technique that builds upon those previously used to measure short-term elevation changes using repeat-track data. This process scheme provides the time series of slope-corrected ice surface heights during each ICESat-2 repeat cycle. Quantifying elevation at different times at the same location is the key to identifying subglacial lakes.

Lake detection in Greenland tends to be complicated by surface lakes. The ATL06 and ATL11 products only capture heights for aggregated photons. If there is a surface lake during data acquisition, the recorded elevation will not be the ice surface but the water surface elevation. This effect can be corrected using the ATL03 data, which provides the depth of shallow water (i.e. reveals the lake bottom ice surface) by its double reflection.

We have added data descriptions in the revised manuscript in the *Data* Section (Page 2, Line 60-61; Page 3, Line 82).

Generally, the results are presented as a range of values across all 51 confirmed lakes. For an ice sheet as large as Greenland, I do not think this is very useful, particularly when the range is across three orders of magnitude (e.g. volume change rates and uncertainties). The lakes are grouped into regions in Figures S5 and S6, so I suggest aggregating the statistics (mean, uncertainties, etc.) by region and discussing how the elevation and volume change rates vary (or do not vary) between regions.

Response:

This is a good idea, thanks. The total lake volume-change rate for each basin has been added as a circle in Figure 2(d), where the circle size is proportional to the magnitude of the absolute rate. We have added the aggregated statistics (mean, uncertainties, etc.) by basin in Figure S5 and S6, and added description in the *Result* Section (Page 8, Line 293-294; Page 8, Line 301-303).

But this can only represent the state of the 61 subglacial lakes that have been detected so far, and it may not be representative of the state of the entire Greenland subglacial system.

I appreciate that the data in the supplementary tables is provided, but I think the spreadsheets would be better placed in an open-sourced repository. The tables in the manuscript could then be used to show aggregate statistics for each region of Greenland assessed.

Response:

    We have uploaded the supplementary tables to the National Tibetan Plateau/Third Pole Environment Data Center.

Other Comments

Abstract: Lake area, height, and volume are an important part of the analysis, so I suggest adding a few lines that mention these parameters.

Response:

    We have added some descriptions on lake characteristics in the *Abstract* Section (Page 1, Line 21-22).

Line 32: "Subglacial lakes can be identified from various remote sensing *techniques*"

  Response: We have changed 'observations' to 'techniques'.

Lines 32-33: What kind of instrument (or instruments) is used to make these inversions?

  Response: Seismic and gravity data can be measured by acoustic impedance or amplitude-versus-angle analysis (Livingstone et al., 2022). We have added this statement in the revised manuscript (Page 2, Line 34-35).

Line 48: Minor nitpick, but the documented footprint size is ~13 m.

  Response: The footprint size is ~11 m according to Magruder et al. (2020) – see also comment by reviewer 2. We have corrected this in the revised manuscript.

Reference:

Magruder, L. A., Brunt, K. M., Neumann, T., Klotz, B., & Alonzo, M. (2020). Passive ground-based optical techniques for monitoring the on-orbit ICESat-2 altimeter

geolocation and footprint diameter. Earth and Space Science. https://doi.org/10.1002/essoar.10504571.1

Line 68: Are all these reference points co-registered with subglacial lakes? If not, I would mention how many reference points are (if that is not too difficult).

Response: This number is the footprints that covered the entire Greenland Ice Sheet, and we counted how many points were within the identified lakes and added them to the *Result* section (Page 7, Line 250).

Line 68: Replace "entire of Greenland" with "Greenland Ice Sheet"

Response: Accept and revised.

Line 88: "a more conservative threshold *of*…"

Response: Accept and revised.

Lines 103-105: Is this a significant problem? Please provide justification on why or why not.

Response: The percentage of lakes that exhibited elevation anomalies during the ArcticDEM period is not a significant problem because it is a preliminary screening step. The lake identification relies on confidence levels determined by the elevation profiles and the long-term elevation change trends.

Line 106: For clarity, this is referring to ICESat-2 tracks, correct?

Response: We added 'ICESat-2' here for clarity.

Line 112: Consider revising to something like: "Supraglacial lakes seasonally form over much of the GrIS ablation zone. These surface lakes may either refreeze on the surface or drain to the ice bed (Selmes et al., 2011)."

Response: Accept and revised.

Line 114: Elaborate on what ATL06 is (i.e. land ice height).

Response: Accept and revised.

Equations 3+4: "confirmred" à "confirmed"

Response: Accept and revised.

Line 168: 2009 à 2019

Response: We rephrased the sentence as 'Spatial patterns of elevation and volume changes over the ICESat-2 period (2019-2020) were generated, and the elevation time-series over the combined ArcticDEM and ICESat-2 periods (2009-2020) were used to determine the temporal patterns of lake activity.'

Line 168: Use "ICESat-2" to prevent confusion.

Response: ArcticDEM was corrected by ICESat data rather than ICESat-2 data, with this information provided in the metadata of the ArcticDEM strip data.

Lines 170-171: Nitpick, but a more up-to-date number is from Brunt et al., (2021), which shows an accuracy of ~0.04 m over ice sheets (reference below).

Response: We corrected this in the revised manuscript.

Lines 177-179: "An additional 10 active lakes were detected by ICESat-2 but not the ArcticDEM" can be removed – the first sentence of the paragraph establishes this already. The rest of the sentence could be rephrased as: "Five of the reported active lakes were missed by ICESat-2, indicating that…"

Response: Accept and revised.

Line 179: "bright" and "strong" are redundant in this context.

Response: We removed these words in the context

Lines 184-185: I am not sure if I understand the connection here. If the lakes with the

most sampling are at upper latitudes, then why would it not be associated with the increased density of ICESat-2 tracks?

Response: We have removed this vague statement in the revised manuscript.

Line 207: Where were these lakes located? A depth of >50 m is awfully large…

Response: These lakes are located in northeastern Greenland, including one known lake in the Flade Isbink Ice Cap. The estimated depth of ~ 60 m for this lake is consistent with the published subglacial lake inventory of Livingstone et al. (2022).

Reference:

Livingstone, S. J., Li, Y., Rutishauser, A., Sanderson, R. J., Winter, K., Mikucki, J. A., Björnsson, H., Bowling, J., Chu, W., Dow, C., Fricker, H., McMillan, M., Ng, F., Ross. N., Siegert, M., Siegfried, M., and Sole, A.: Subglacial lakes and their changing role in a warming climate. Nat. Rev. Earth Environ., 19, https://doi.org/10.1038/s43017-021-00246-9, 2022.

Lines 211-212: "Lakes with both positive and negative elevation change rates during the study period were found in each basin with few exceptions." Redundant sentence – consider revising or removing.

Response: We removed the sentence in the revised manuscript.

Line 238: Terms such as "quiescence" and "high stand" are infrequently used jargon. I suggest using other terminology (or define what "high stand" means) to make it more understandable to a general audience.

Response: We defined this pattern in plain language in the revised manuscript (Page 9, Line 332).

Line 241: "The temporal resolution of the ArcticDEM varies[,]…" (add comma)

Response: Accept and revised.

Conclusions: This needs a paragraph on how this paper would benefit future research

studies, or how future studies could build upon the limitations or difficulties discussed in this study.

Response: We added a paragraph about the limitations and difficulties (i.e., the discrimination between surface and subglacial lakes), and how further studies can build on the quantification of surface water on subglacial water systems.

Figure 1: A colorbar label (with units) is needed.

Response: The label and units has been added in Figure 1.

Lines 381-382: "The red-blue lines represent the differences between ICESat-2 tracks…"

Line 382: "…while *the* grayscale colorbar is the ArcticDEM."

Lines 382-383: "Elevation anomaly profiles across the subglacial lake are given for RGT 321 (b) and RGT 162 (c)."

Response: We rephrased the figure caption as follows 'The red-blue line represents the elevation change rate derived from the linear fit of ICESat-2, while the grayscale colourbar represents the elevation change rate derived from the ArcticDEM. Elevation anomaly profiles across the subglacial lake are given for RGT 162 pt3 (b) and RGT 321 pt2 (c).'.

Line 384: "Their spatial locations are also indicated in (a)." Redundant sentence.

Response: We removed the sentence in the revised manuscript.

Figure 2: I suggest adding a label for the colored circle legend in (b), otherwise I am not sure what the colors represent (or what the units are).

Response: We added the legend in this Figure. The colored circles in Figure 2(b) indicates the percentage of lakes that belong to this grade of area.

Figure 3: Please provide a legend for the differently colored dots.

Response: We added this legend in the revised manuscript.

---

## Author Comment (AC2)

Dear Editor and Reviewer,

We would like to thank you for your detailed feedback, and we address each of their specific comments below. However, we would like to address up-front the major criticism that the elevation anomalies we observe are surface lakes and not subglacial lakes.

The fact many of the locations are associated with surface water does not preclude a subglacial lake as the driver of elevation changes. Indeed, we would expect this as subglacial lake drainage will create an ice-surface depression that would then naturally at least partially fill with surface water. Clearly where there is no surface water, we can have more confidence in a subglacial lake origin. However, where this is not the case, we believe we can still identify subglacial lakes via the following criteria.

- Assessing the ice surface profiles (ICESat-2&ArcticDEM). We can use this approach to identify wavy (i.e. elevation profiles that could not represent surface water or surface water covered with an ice lid) profiles that are not associated with surface water (e.g., Figure 1 of the main manuscript) but nevertheless still display an elevation change. We also provided more examples below (Figure R1). Sequence of ArcticDEM images at the same location may also confirm the elevation anomaly. Where surface water is identified we can evaluate whether the extent of surface water is the same size as the areal extent of the elevation change anomaly. Where the surface water is shallow (<7 m) we can also use ICESat-2 to measure the height of the lake bottom, which would be independent of surface water.

[Figure]

Figure R1 Elevation anomaly profiles across three subglacial lakes for RGT 963 pt2 (a), 986 pt1 (b), and 1276 pt3 (c).

- Investigating the temporal pattern and magnitude of elevation change. Surface lakes

typically show a sub-seasonal pattern of filling and draining and experience elevation changes of <10 m. In contrast previously identified active subglacial lakes in Antarctica, Greenland and Iceland show multi-year fill-drain behaviours (Livingstone et al., 2022). We individually analysed each potential lake to determine the elevation change pattern. All of our medium and high confidence lakes displayed multi-year patterns, often with elevation change magnitudes of >10 m.

- Finally, we adopted a confidence level for each potential lake based on the above analysis – low, medium and high – with all low confidence lakes removed from the inventory.

Overall, we present multiple lines of evidence to support our interpretation of active subglacial lakes in the ablation zone of the Greenland Ice Sheet, and develop a method that will allow others to deal with the challenge of discriminating subglacial and surface lakes. We recognize there is some uncertainty (as captured in our medium and high confidence levels), but this study represents a major step-forward in demonstrating that these lakes do exist and are widespread.

On behalf of all the authors,

Yubin Fan

Reviewer #2

I really don't think the main premise of this study is correct. The authors have found 61 locations on Greenland where ICESat-2 and ArcticDEM strips indicate that the ice-sheet surface height has changed, and they take these changes to indicate the presence of subglacial lakes, in some cases arguing that the temporal pattern of height change is diagnostic of subglacial lake activity. I think the simplest assumption, that needs to be carefully considered, and requires strong evidence to be disproven, is that the height changes observed here are the result of supraglacial lake activity.

I did a spot check of the locations in table S2 against Google Earth, and, of the locations for which Google Earth had high-resolution imagery available (i.e. most of the west-coast lakes, and most in the far northeast), every single one had at least some sign of a supraglacial lake, although most were not identified in column I as being twinned with a supraglacial lake. In some cases, the lake was partially snow covered, and in others it was drained at the time of the Google Earth imagery, but it seems plausible that all of the observed height changes were a result of supraglacial water motion. It appears that the authors only identify lakes as being associated with a supraglacial lake when they can clearly see a double surface in the ICESat-2 data, but inspection of visible imagery is an easy way to see whether a particular location is in a lake basin, and it's often possible to see water at the surface.

Supraglacial lakes are known to be common in the ablation areas of Greenland. They can fill and drain in a single season, or can be present over multiple years, freezing partially or completely in the winter, and often have lids of frozen lake and snow that may or may not melt in the summer, and whose height can vary over time as water flows into or out of the lake (for example, the lake in figure S1 appears to be partially covered with frozen lake ice). As a result, height changes at the surface of the ice sheet may reflect changes in supraglacial lake volume, even when water is not visible at the surface. Because of this, any height-change within a lake basin should be suspected to be as a result of supraglacial processes. These can include not just

changes in the level of exposed water, but also, changes in water level below a floating layer of lake ice, or enhanced ablation due to water flow or due to the presence of low-albedo sediments in the lake basin.

Response:

We believe that our premise is right (see also our general comment at the beginning of this reply). Certainly, supraglacial lakes are common within the ablation zone of the Greenland Ice Sheet. But, recent work has demonstrated that active subglacial lakes also exist in this zone (Livingstone et al., 2022 and references herein). Subglacial and supraglacial lakes are likely to be linked as whenever a subglacial lake drains it will create a surface depression (as also seen in Iceland and Antarctica) that could then fill with water creating a surface lake. We recognize this in the manuscript and have devised an approach to discriminate between the two lake types ("3.3 Impact of surface lakes on the detection of subglacial lakes") that accounts for uncertainty in our results ("3.4 Lake confidence level classification").

From analysing the elevation change data, we suggest the elevation-change pattern is demonstrably different between the two lake types. In particular, surface lakes produce a flat (or broadly horizontal with small undulations if there is a floating ice lid on top of water) surface elevation profile and typically show seasonal elevation change patterns. We used three criteria to discriminate confidence levels, (1) whether the elevation anomaly has a flat surface on the elevation profile (Table S2, Column 'Flat spot?'), (2) whether the elevation change pattern is characteristic of long-term subglacial lake drainage or filling (Table S2, Column 'Timeseries conf'), (3) Elevation anomaly profiles across the subglacial lake for each RGT and its magnitude of elevation change (Table S2, Column 'ATL conf'). The three separate confidence levels are integrated into a total confidence level (as stated in Section 3.4). Potential lake locations that were classified as low-confidence were not listed and discussed in the paper. The 15 high confidence subglacial lakes we identified typically exhibited large elevation changes over multi-year periods and with bumpy ice surfaces indicating that at least some of the elevation change was not associated with surface water.

Subglacial lakes affected by surface lakes that we identify in the table are those lakes with an obvious double-layer reflection in the ATL03 profile (the measurement time of ICESat-2). To further verify this, we checked the Landsat imagery around the acquisition time of ICESat-2 to further confirm the existence of surface water (e.g., Figure R2). In addition, surface water does not necessarily mean there is not also a subglacial lake; as we state once a subglacial lake drains it is not surprising that a surface lake occurs in the collapsed ice-surface basin (Willis et al., 2015).

[Figure]

Figure R2 Manual check on surface water around the ICESat-2 acquisition time, taking Lake ACADEMY05 as an example. ICESat-2 shows an elevation anomaly in 04 September 2019, and both ATL03 data and LandSat-8 image (04 September 2019) confirm that there is very little surface water (affect only one ICESat-2 footprint) that day.

For a few of the lakes, water is obviously present in the ICESat-2 data, and the authors have attempted to use a technique based on ATL03 photon data to measure changes in the lake-bed height at times when the lakes are ice filled.  This technique (the Watta algorithm) involves estimating the depth of the lake water based on the returns from the

surface and bottom of the lake. There are two problems with the current study's estimates of ice-surface height (i.e. lake-bottom height) based on these results. First, the return form the bottom is typically quite diffuse, so that the photons used for measuring the lake bottom can come from a range of depths below the lake bottom itself. This leads to an uncertainty in the depth that is not quantified here; because lighting conditions, sediment load on the bottom of the lake, and the scattering characteristics of the lake bottom could all influence the diffuseness of the bottom reflector, it seems likely that uncertainty in identifying the height of the lake bottom in the ATL03 data could lead to substantial scatter in estimates of lake-bottom height and thus to apparent height and volume change where there is none. Second, the authors use the lake-bottom heights as calculated from (presumably) single ICESat-2 beams to replace the heights in the ATL11 time series (see my comment on line 124) without using the ATL11 polynomial surface to correct for the position of the measurements relative to the ATL11 reference point. This can lead to potentially large inconsistencies between the ATL11 time series and the heights from the Watta algorithm.

If the supraglacial lakes are paired with subglacial lakes, there should be a diffuse lake-like signal that extends outside the supraglacial lake boundary. It might sometimes be possible to see the water that leaves draining supraglacial lakes as it inflates the subglacial water system. However, in the few examples where this kind of behavior has been observed using GPS (see Das et al, 2008), the uplift was so brief that it would require considerable luck to observe it with ICESat-2 or with a Worldview DEM.

Response:

We acknowledge these uncertainties, but believe they have a limited effect on lake determination and its elevation/volume change estimation. First, ICESat-2 cannot identify all surface lakes because there is a depth limit (~7 m, Fair et al., 2020), and the estimated depths derived from the Watta algorithm have a high correlation with the image-based depths derived from Landsat-8/Sentinel-2 and manually-picked photon estimates (Fricker et al., 2020). Despite having an uncertainty in the depth estimation,

the corrected elevation can provide information on whether there remains a residual (i.e. subglacially-derived) elevation anomaly. The spatial offset between ATL03 acquisition location and ATL11 RPT is rather small across the GrIS. The elevation derived from ATL03 and ATL11 have a mean difference of about – 0.17 m for the corrected 18 lakes, and this value is considerably smaller than the detected elevation changes (as shown in Table S3, generally >5 m). Thus, this correction does not significantly affect the change trend, and is within the calculated elevation-change uncertainty.

Reference:

Fair, Z., Flanner, M., Brunt, K. M., Fricker, H. A., and Gardner, A.: Using ICESat-2 and Operation IceBridge altimetry for supraglacial lake depth retrievals. The Cryosphere, 14, 4253–4263, https://doi.org/10.5194/tc-14-4253-2020, 2020.

Fricker, H. A., Arndt, P., Brunt, K. M., Datta., R. T., Fair, Z. and Jasinski, M. F.: ICESat-2 Meltwater Depth Estimates: Application to Surface Melt on Amery Ice Shelf, East Antarctica. Geophys. Res. Lett., 48, e2020GL090550, https://doi.org/10.1029/2020GL090550, 2020.

Comments on specific lines within the paper follow.

Line 47: "ICESat-2 has an improved footprint size (approximately 17 m with 0.7 m along-track)." The authors should be clear that the footprint size is 11 m (not 17, see Magruder et al, 2020) and the along-track pulse-to-pulse spacing is 0.7 m.

Response:

We corrected it in the revised manuscript.

Line 90: "The elevation-change anomaly associated with subglacial lake filling and/ or drainage should have a characteristic spatial pattern comprising an obvious elevation anomaly at the lake center while the outside remains stable." I don't think this is necessarily true. For a lot of subglacial lakes, the height-change signal is fairly

spatially diffuse, with smooth variations in the height-change signal reflecting the flexure of the ice. It seems reasonable for subglacial-lake-driven height changes to be smooth at a scale of around one ice thickness. Truly sharp spatial patterns in height change are more likely associated with supraglacial lakes.

Response:

We respectfully disagree that there will always be a smooth transition. Where ice is thinner the elevation change signal might produce a more clearly defined boundary. This is demonstrated for some Antarctic subglacial lakes in the Antarctic Peninsula, which have produced sharp elevation changes (e.g., Hodgson et al., 2022, Figure 7). In contrast, over thicker ice (typical of subglacial lakes in Antarctica) the transition to no elevation change is more gradual. Therefore, the elevation changes associated with active subglacial lakes in Greenland may be not as smooth as those identified on the Antarctic Ice Sheet.

We have rephrased this part as follows 'The relative elevation-change anomaly associated with subglacial lake should have a characteristic spatial pattern comprising an obvious elevation anomaly at the lake center which reduces to zero (within uncertainty) outside the lake. '.

Reference: Hodgson, D. A., Jordan, T. A., Riley, T. R., and Fretwell, P. T.: Drainage and refill of an Antarctic Peninsula subglacial lake reveals an active subglacial hydrological network, The Cryosphere Discuss. [preprint], https://doi.org/10.5194/tc-2022-144, in review, 2022.

Line 100: "Subglacial lakes which also coincided with elevation anomalies during the ArcticDEM period were confirmed as subglacial lakes". Is this the definition of "confirmed" and "unconfirmed?" I don't think this is a good way to confirm that a given lake is subglacial rather than supraglacial—ArcticDEM data can show height changes associated with changes in lake ice on top of supraglacial lakes, and can show height differences between filled supraglacial lakes with frozen lake ice and drained

supraglacial lakes. Similar problems can be seen in comparisons between ArcticDEM and ICESat-2 data.

Response:

ArcticDEM cross-validation is mainly used to determine whether there is an elevation anomaly. "Confirmed" in the text does not necessarily mean that this location is a subglacial lake. Instead it indicates that these locations are more likely to be subglacial lakes. However, only the long-term elevation time-series of these locations was subsequently discriminated. We have now specified that these are potential subglacial lakes in the revised manuscript (Page 4, Line 122-123).

Line 124: "The bottom height was taken as the corrected ATL11 elevation"—I take this to mean that the bottom height calculated based on the Watta algorithm was used in place of the elevation from ATL11 for that cycle. Simply replacing an ATL11 value with a single-beam elevation from ATL03 risks mixing values that have had different corrections applied. ATL11 heights are calculated relative to the ATL11 reference surface, which takes into account the local shape of the ice-sheet surface. The appropriate way to do this calculation would be to use the ATL11 polynomial coefficient fields to calculate the height of the reference surface at the location of the location of the Watta height estimate, and subtract the reference surface height from the Watta height.

Response:

The spatial offset between the ATL03 acquisition location and ATL11 RPT is small for the GrIS. The elevations derived from ATL03 and ATL11 have a mean difference of about $-0.17$ m. This value is smaller than the detected elevation changes for each difference between two lake heights (as shown in Table S3, generally >5 m), so this correction does not substantially affect the change trend of this lake.

Line 125 (figure S1). As noted in my comment about line 90 (above), the surface-height changes associated with subglacial lakes in Antarctica are spatially smooth, which is expected based on the mechanics of ice deformation. Although the authors

do not plot the height change they would estimate between 2019 and 2020, the difference between the lines in figure S1b indicates that it would be fairly jagged. This suggests to me that at least part of the signal visible in figure S1 is either due to errors in the bottom-height estimate from the Watta algorithm or due to melt.

Response:

Figure S1b shows the elevation profile before the Watta correction. When we plot the corrected elevation profile (Figure R3), it shows that this position still maintains the elevation anomaly (>5 m), which is rather large compared to the depth uncertainty (with a mean absolute difference of 0.33 m compared with manual-pick depths, derived from the supplement table of Fricker et al. (2020)). We have added these statements in the revised manuscript (Page 5, Line 160-161).

[Figure]

Figure R3 Elevation anomaly profile across the subglacial lake along the ICESat-2 RGT 0582 pt2 after Watta correction.

Section 3.4 (129-136): I don't see a good reason to believe that subglacial lakes on Greenland should show the "multi-year pattern of filling and then rapid drainage" that the authors take as indicative of a subglacial lake. Supraglacial water inputs to the bed of the Greenland ice sheet are large and seasonal, which could quickly fill or overfill

subglacial lakes, leading to filling and drainage on subseasonal timescales.

Response:

Multi-year patterns of filling and then rapid drainage have already been identified for other active subglacial lakes in Greenland (e.g., Figure R4) (Willis et al., 2015; Bowling et al., 2019; Livingstone et al., 2019; Liang et al., 2022). This pattern suggests that despite large water inputs to the bed, subglacial lakes do not always quickly fill and then drain on sub-seasonal timescales (although the reviewer is right that this probably also happens; but such instances are then more difficult to distinguish from typical seasonal supraglacial lake behaviour). A key finding of this paper is that this multi-year style of behaviour is widespread in the ablation zone.

Reference:

Willis, M. J., Herried, B. G., Bevis, M. G., and Bell, R. E.: Recharge of a subglacial lake by surface meltwater in northeast Greenland. Nature, 518(7538), 223-U165, https://doi.org/10.1038/nature14116, 2015.

Liang, Q., Xiao, W., Howat, I., Cheng, X., Hui, F., Chen, Z., Jiang, M., and Zheng, L.: Filling and drainage of a subglacial lake beneath the Flade Isblink ice cap, northeast Greenland. The Cryosphere Discussions, pp.1-17, https://doi.org/10.5194/tc-2021-374, 2022.

[Figure]

Figure R4 Elevation time-series over the combined ArcticDEM and ICESat-2 periods (2009-2020) within the lake and the lake buffer (up panel), and its corresponding elevation anomaly (down panel), taking Lake ICE_CAPS_NE01 as an example. Note the pattern of multi-year filling and then rapid drainage.

In contrast to subglacial lakes, the elevation change caused by supraglacial water tended to show a pattern of seasonal change (Figure R5); these elevation anomaly patterns are removed from the lake inventory.

[Figure]

Figure R5 Elevation time-series over the combined ArcticDEM and ICESat-2 periods (2009-2020) of one surface lake, which shows a pattern (where data frequency allows) of seasonal elevation change without a significant long-term trend.

Section 3.5. I don't see a description of how height anomalies are calculated. In figure 3 and in table S4, some of the lakes have anomalies that are always negative or always positive. To what are these anomalies relative?

Response:

The elevation anomalies in table S3/S4 and Figure 2/3 are relative values, which were calculated by subtracting the median ice-surface elevation within the region overlying the subglacial lake from the buffer around it.

We rephrased this part as follows 'The elevation-change rate within the lake

polygons is composed of ice-flux divergence, ice ablation and water motion (Smith et al., 2009), while the ice outside is only affected by ice-flux divergence and ablation. This 'background' elevation change needs to be subtracted to calculate the relative elevation-change caused by the subglacial lake. For each ICESat-2 overpass, we first calculated the median value of all ICESat-2 measurement points within the lake polygon, and then the median elevation of the area surrounding the lake (within the buffer-region) was subtracted to produce the elevation anomaly.'.

Reference:

Smith, B., Fricker, H., Joughin, I., and Tulaczyk, S.: An inventory of active subglacial lakes in Antarctica detected by ICESat (2003–2008). J. Glaciol., 55(192), 573–595, https://doi.org/10.3189/002214309789470879, 2009.

Section 3.6: This section should come before section 3.2, where the results of the ArcticDEM strip registration are used to try to confirm the lake locations.

Response:

We moved this to Section 3.2.

Line 145: what does it mean that "the effect of the buffer size on the elevation-change rate was neglected"? Does this mean that the authors do not account for the effect of the buffer size in their uncertainty calculations? Or does it mean something else?

Response:

What we meant here is that the effects of different buffer sizes (the radius of a circle whose area is equal to the lake and a buffer of half the radius of the standard buffer in this paper) on the elevation-change rate estimations was small (~6%). In addition, buffer selection does not affect temporal lake trends (i.e., fill or drain), so we neglect the effect of the buffer size on our lake elevation change data.

We used the radius buffer because this produced a similar footprint number to the lake region providing a more robust estimation.

Line 177: "missed by ICESat-2"—do the authors mean that the lakes locations were sampled by ICESat-2, but no height anomaly was detected? This would be typical for subglacial lakes in Antarctica, which often show episodic activity.

Response:

Two active lake locations found by Bowling et al. (2019) and a lake found by Livingstone et al. (2019) (67.178°N, 50.149°W) were all sampled by ICESat-2. Although elevation anomalies (> ±2 m) were detected, these locations have been removed (e.g., Figure R6, RGT 1169 pt2, sample Lake Isunguata Sermia 2) because they did not show a characteristic temporal pattern of subglacial lake filling and draining during the ICESat-2 period.

[Figure]

Figure R6 Elevation anomaly profiles across the subglacial lake are given for RGT 1169 pt2 (a) and the corresponding elevation-change rate (b), indicating no constant elevation change of this known subglacial lake.

Line 179: "No classic bright, flat and strong reflections were found from analysis of RES data." How many of the lake locations were directly sampled by RES surveys?

I'd be surprised if all 61 were.    Please specify which were and were not sampled.

Response:

Subglacial lakes derived from RES data during 1993-2016 have been published by Bowling et al. (2019) and Livingstone et al. (2022). These 57 stable lake locations were sampled by ICESat-2, but no height anomaly was detected (Figure R7, Lake located in Kangerlussuaq Gletsjer, tally:44 in the Livingstone inventory).

[Figure]

Figure R7 Elevation anomaly profiles across one stable subglacial lake are given for RGT 1169 pt2 (a) and the corresponding elevation-change rate (b), indicating this lake shows stable elevation.

We only checked the RES data from 2017-2019, and 10 of 50 confirmed lakes (for which we have a lake boundary) located in the southwestern GrIS were sampled by RES data, and no flat reflections were found (Figure R8).

[Figure]

Figure R8 Relative basal reflectivity (a), the bed elevation (b), and hydraulic potential of one active subglacial lake detected by ICESat-2.

We rephrased this part as follows 'RES data during 1993-2016 were analysed by Bowling et al. (2019), revealing 57 stable lakes. Of the 57 stable lakes, 39 of them were sampled by the ICESat-2 ATL11 data (within a circular buffer using the 1/2 length provided by Livingstone et al. (2022)), but no clear elevation anomaly was found. In addition, 10 of the 61 active lakes were sampled by RES data from 2017 to 2019, but no classic flat reflections were found.'.

Line 184: " However, since lakes occur at all latitudes, we infer that their occurrence has no connection with the spatial density of ICESat-2 tracks."   Obviously, the occurrence of subglacial lakes has no connection with the spatial density of ICESat-2 tracks.   The detection of the lakes, on the other hand, almost certainly does, because many of the lakes are small compared to the track-to-track spacing of ICESat-2 data. It seems likely that at low latitudes, ICESat-2 misses many more lakes than it does at

high latitudes.

Response:

We have removed this vague statement in the revised manuscript.

187-197—the simplest explanation for the association between the lakes in this study, negative surface mass balance, and thin snow cover, is that the lakes observed here are supraglacial, not subglacial.

Response:

We respectfully disagree (see response to major comment above). The lakes' distribution agrees with previously published data (Livingstone et al., 2022), and predictions by Bowling et al. (2019). Thick firn limits the amount of surface-derived water that reaches the ice bed in the southeastern and inland sectors of Greenland. Lakes located in these regions get limited or no surface recharge and therefore their elevation changes are small and cannot be captured by the altimetric data

198-204: the sizes of the lakes are also consistent with supraglacial, not subglacial lakes.

Response:

Steeper ice-surface slopes and thus subglacial hydrologic gradients control the morphology of subglacial lakes in Greenland (Bowling et al., 2019). Therefore, the size of the lakes underneath the Greenland Ice Sheet is typically smaller than those in Antarctica. Indeed, the lakes we identify are similar to other active lakes identified in Greenland (Bowling et al., 2019; Livingstone et al., 2019) and in Iceland (Livingstone et al., 2022), with typical areas of 0-10 $km^2$, and lengths mostly between 1-2 km.

Reference:

Bowling, J.S., Livingstone, S. J., Sole, A. J. and Chu, W.: Distribution and dynamics of Greenland subglacial lakes. Nature Communications, 10:2810, https://doi.org/10.1038/s41467-019-10821-w, 2019.

Livingstone, S. J., Sole, A. J., Storrar, R. D., Harrison, D., Ross, N., and Bowling, J.: Brief communication: Subglacial lake drainage beneath Isunguata Sermia,West

Greenland: geomorphic and ice dynamic effects. The Cryosphere, 13, 2789–2796, https://doi.org/10.5194/tc-13-2789-2019, 2019.

Livingstone, S. J., Li, Y., Rutishauser, A., Sanderson, R. J., Winter, K., Mikucki, J. A., Björnsson, H., Bowling, J., Chu, W., Dow, C., Fricker, H., McMillan, M., Ng, F., Ross. N., Siegert, M., Siegfried, M., and Sole, A.: Subglacial lakes and their changing role in a warming climate. Nat. Rev. Earth Environ., 19, https://doi.org/10.1038/s43017-021-00246-9, 2022.

---

## Referee Report (RR1)

The authors have given a revised manuscript detailing the detection and monitoring of subglacial lakes using a combination of ICESat-2 data and the ArcticDEM. Overall, the manuscript is improved from its previous iteration, with text and figures that are easier to understand. I do have a few suggestions and clarifying questions that I would like to see addressed before it is ready for publication:

**Page 3, Line 67:** Small nitpick, but I suggest being specific here and noting that ATL06 measures land ice height.

**Page 3, Line 79:** Since you are using ATL03 to identify supraglacial lakes, I suggest giving the full name of the product (Geolocated Photon Data) and giving a bit more detail on what is in ATL03 data.

**Page 3, Line 88:** Just to make sure, these published subglacial lakes were found using the ArcticDEM?

**Page 4, Lines 103-104:** How was it determined if other factors caused the elevation anomalies? I imagine that it could be difficult to distinguish between lakes and rough topography.

**Page 4, Line 107:** ICESat-2, not ICESat. Also, what exactly was corrected from the DSMs, and using what metadata?

**Page 5, Line 134:** If you define ATL06 on Page 3, Line 67, then it will not be needed here.

**Page 6, Line 170**: 2 km (spacing)

**Figure 1:** I am assuming that "pt" refers to the pair tracks, but what do the numbers indicate?

**Figure 3:** I notice that stable lakes are generally found either on the eastern part of the ice sheet or on the northern margin. Is this a coincidence? I would like to see the authors' interpretation.

**Figure S3:** It is interesting that a large lake was found, but I am not sure what unique information is provided by this figure. Was the drainage rate (or lack thereof, looking at 2013-2017) surprising for a lake that large? If not, then I would consider removing this figure.

**Figure S4:** There is a caption here, but no figure. Is the figure missing, or was it removed?

---

## Author Response (AR2)

Dear Editor and Reviewer,

We would like to thank you for your detailed feedback. We address each of the specific comments below.

The major issue with this manuscript raised by reviewer is the differentiation between subglacial lakes and supraglacial lakes and the processing scheme of the ArcticDEM strips. Considering the previous suggestions, here we summarize the major changes to the revised manuscript:

1) ArcticDEM processing: In this iteration, we only used the strips that provided the co-registration information against ICESat to ensure data consistency, and excluded the strips without this information. In addition, we produced ArcticDEM data by only using the pixels that overlapped with the ICESat-2 ATL11 data to construct a time series of lake elevation profiles and its corresponding elevation anomaly. We removed the flat spots in the ArcticDEM period where the elevation profiles could not be corrected. This DSM quality control leads to fewer DSM tiles, and 6 lakes were not covered by high-quality DSM so were classified as 'unconfirmed lakes' (Table S2). However, we still have two datasets to distinguish subglacial lakes and supraglacial lakes, including elevation profiles during 2009-2020 (ICESat-2 period), and the combined elevation anomaly during 2009-2020 (see all our data for each lake provided in Figures S4 to S21).

2) Differentiation between subglacial lakes and supraglacial lakes: We adapted our criterion from large elevation changes to no abrupt elevation changes because supraglacial lakes are characterized by a seasonal fill-drain pattern, whereas subglacial lakes tend to fill over multiple year. For flat surfaces, we used the Watta algorithm to correct the water bottom during the ICESat-2 period, and removed flat spots (which cannot be corrected) during the ArcticDEM period. The addition of elevation profiles from the ArcticDEM period increased the data volume, and therefore the confidence level for discriminating supraglacial lakes. We removed lakes that exhibited sudden elevation changes or that were mainly characterized by flat surfaces, and only retained lakes with profiles exhibiting characteristic patterns like Figure 1 b/c. After this lake confidence-level re-classification, 18 lakes with high/medium confidence level were retained.

Overall, our interpretation of active subglacial lakes of the Greenland Ice Sheet combines ArcticDEM and ICESat-2 to deal with the challenge of discriminating subglacial and supraglacial lakes. We recognize there is some uncertainty (as captured in our medium and high confidence levels), but this study represents a step-forward with the addition of 16 new active lakes (18 in total).

On behalf of all the authors,
Yubin Fan

Reviewer #1

The authors have given a revised manuscript detailing the detection and monitoring of subglacial lakes using a combination of ICESat-2 data and the ArcticDEM. Overall, the manuscript is improved from its previous iteration, with text and figures that are easier to understand. I do have a few suggestions and clarifying questions that I would like to see addressed before it is ready for publication:

We thank the reviewer for the helpful feedback, we are appreciative of his or her help and time.

Page 3, Line 67: Small nitpick, but I suggest being specific here and noting that ATL06 measures land ice height.

Response:
   We have specified 'land ice elevations' here.

Page 3, Line 79: Since you are using ATL03 to identify supraglacial lakes, I suggest giving the full name of the product (Geolocated Photon Data) and giving a bit more detail on what is in ATL03 data.

Response:
   We have added the full name and some details of the ATL03 products here.

Page 3, Line 88: Just to make sure, these published subglacial lakes were found using the ArcticDEM?

Response:
   Known active subglacial lakes were detected by ArcticDEM ice-surface elevation change, and the stable lakes were detected by RES data. We have added the information in the revised manuscript (Page 3, Lin 90-91).

Page 4, Lines 103-104: How was it determined if other factors caused the elevation anomalies? I imagine that it could be difficult to distinguish between lakes and rough topography.

Response:
   Displacement of the ICESat-2 tracks has been corrected by ATL11 product, and the slope generated by the mosaicked 100-m ArcticDEM product was used as a topographic reference.

Page 4, Line 107: ICESat-2, not ICESat. Also, what exactly was corrected from the DSMs, and using what metadata?

Response:
   The correction parameters have been provided in the metadata of each DSM strip, and the offsets were obtained by the co-registration between each DSM and ICESat. These values were not calculated by this paper.
   We rephrased the sentence as follows 'We only used DSM strips where correction vectors

obtained by the co-registration between filtered ICESat altimetry data were provided within the metadata. (Page 4, Line 107-109)'

Page 5, Line 134: If you define ATL06 on Page 3, Line 67, then it will not be needed here.

Response:
   We have moved the name of ATL06 product to the Data section.

Page 6, Line 170: 2 km (spacing)

Response:
   Accept and revised.

Figure 1: I am assuming that "pt" refers to the pair tracks, but what do the numbers indicate?

Response:
   The numbers mean the order of pair tracks. We added the explanation here.

Figure 3: I notice that stable lakes are generally found either on the eastern part of the ice sheet or on the northern margin. Is this a coincidence? I would like to see the authors' interpretation.

Response:
   We assumed that it was caused by the different ice thickness and different surface mass balances. High accumulation rates and thick firn limit the amount of surface-derived water that reaches the ice bed in the eastern part of the ice sheet, so the water budget of lakes in these regions are hard to change. We have explained the reason in the manuscript (Page 8, Line 220-221, Line 225).

Figure S3: It is interesting that a large lake was found, but I am not sure what unique information is provided by this figure. Was the drainage rate (or lack thereof, looking at 2013-2017) surprising for a lake that large? If not, then I would consider removing this figure.

Response:
   We have removed this figure in the supplementary information.

Figure S4: There is a caption here, but no figure. Is the figure missing, or was it removed?

Response:
   This figure has been provided in the supplementary information.

Reviewer #2

The authors are making progress on this study and have fixed some of the material that I objected to in the previous versions. I'm still not sure about what remains, and I think the authors need to spend some time thinking about the quality evidence that they have presented, and whether they, as referees of an article by a different set of authors, would be persuaded by the arguments and data they present.

We thank the reviewer for the helpful feedback, we are appreciative of his or her help and time.

I remain skeptical of the authors' differentiation between subglacial lakes and supraglacial lakes. The authors claim that the use of ATL03 and the Watta algorithm lets them detect elevation changes even when there is water in the lake, but they don't make any distinction in the text or in the tables as to which elevation differences were calculated using the ATL03/Watta method. The one example they show of ATL11 and ATL03/Watta is in figure S1, which is not discussed in enough detail for me to be able to understand how they used the data, or what they thought was happening in that example. I had to cross-reference the coordinates on the figure with table S2 to figure out that this was HAYES_GLETSCHER_N_NN01. Then, looking at table S4, I could see that the authors listed elevation anomalies that almost certainly were measured on the floating ice on top of the lake alongside the elevation anomalies measured when the supraglacial lake may have been empty, and alongside the ATL03/Watta anomalies from August 2019.

Response:
    We first added a column 'Watta correction?' in Table S2 to distinguish the lakes corrected by the Watta method. The Watta algorithm does not only estimate depth estimates, but can also provide lake characteristics (the presence of refrozen ice at the surface) (Datta et al., 2021). For each potential lake, we manually checked the elevation profiles and identified the flat surface. We recorded the data acquisition date and downloaded the corresponding ATL03 photon data.
    We used a Figure to show how ICESat-2 can penetrate through the water column of supraglacial lakes to measure the lake bottom (Figure 2 in the manuscript). We used example without an ice lid here (Lake NIOGHALVFJERDSFJORDEN01) in the new Figure S1.

The use of the ATL03/Watta algorithm is not possible for the ArcticDEM data. This means that the time series of elevations from mid-2018 and earlier are likely measuring changes in supraglacial water, or in lake ice atop supraglacial water. Unless the authors present good evidence to the contrary, this should be the assumption for what is going on in this earlier part of the record. As a result, examination of the ArcticDEM record does not confirm that the elevation anomalies are subglacial lakes- it just confirms that there is elevation variation in the past that continues into the ICESat-2 period.

Response:
    It is true that the ArcticDEM record is not direct evidence of subglacial lake activity and we have added a sentence to state this effect (Page 4, Line 121-122). However, the extension of the elevation change record can give us a more comprehensive picture of the patterns of elevation changes (e.g., whether the elevation changes were abrupt), which was critical for discriminating

subglacial lakes from other processes. In addition, we can also identify how many time periods have flat surfaces for one lake.

In their rebuttal the authors show three examples where they claim that the irregular surfaces demonstrate that they are not measuring supraglacial lakes. I don't understand why they make this claim—the first example (figure R1a) very clearly shows a supraglacial lake that has filled and drained seasonally. When water is present, the surface is not perfectly flat, but the irregularities could easily be caused by a rough lid of floating ice, or by spatially variable laser-light penetration into water and/or detector saturation effects from a bright water reflection.

Response:
    Figure R1 (a) may be a misinterpretation of a subglacial lake signal. We adapted our criterion from big elevation changes to no abrupt elevation changes over time. We reclassified the lake confidence level and removed any lakes that look like Figure R1 (a) (i.e., contains abrupt elevation change) from our dataset.

    Figures R1b and R1c are much better examples of potential subglacial lake activity, but having seen the authors misinterpretation figure R1a, I am very worried about the quality of the interpretations of other data in the manuscript, and I would encourage the editor to request that the authors present the height profiles interpreted in the study one-by-one in the supplemental material to a revised manuscript. This would give the authors the opportunity to present the ATL11 tracks with the corresponding points sampled from ArcticDEM to provide a long-term record of change for each of the lakes. I suspect that in many cases, this would show that the ArcticDEM data sample flat surfaces (i.e. supraglacial water) during the high stands for many of the lakes.

Response:
    We have recalculated the mean elevation within the lake and its buffer, and the corresponding elevation anomaly during the ArcticDEM period (provided in Table S4) by only using pixels sampled by ATL11 data. We further provided the elevation profiles in Figures S4 to S21.

The authors claim to have looked at Landsat imagery to confirm or deny the presence of liquid water in the lakes and, in their rebuttal, show an image for Academy05 that does not show much water on the surface. However, at the time of this image, Academy05 was at a low stand relative to the ICESat-2 measurements, so if it is a subglacial lake, we wouldn't necessarily expect to see water at this time. Further, the landsat data can't rule out floating ice for the lake.

Response:
    We agree that we cannot rule it out from the Landsat imagery alone. Academy05 and other lakes showing the same case were eliminated in this version because they did not show gradual elevation change.

The method for combining ArcticDEM and ICESat-2 time series seems to have the potential to

generate nonsensical time series. The ICESat-2 profiles sample a small part of each lake basin, while the ArcticDEM data sample the whole basin, which means that a nonuniform pattern of filling and drainage (subglacial or supraglacial) will produce different values for the two datasets that are likely not comparable. I suggest sampling the ArcticDEM DEMs at the locations of the ATL11 measurements to construct a self-consistent time series, and investigating the extent to which these self-consistent time series agree with the full-basin records derived from ArcticDEM. If they don't agree, the two should be presented separately, not combined as they currently are.

Response:

    We agree that the different sampling between ICESat-2 and ArcticDEM may lead to bias in the long-term elevation series. We recalculated the mean elevation within the lake and its buffer, and the corresponding elevation anomaly during the ArcticDEM period (as provided in Table S4) by only using pixels that can be sampled by ATL11 data (Page 5, Line 130-133). We further extend the elevation profiles along the ICESat-2 track to 2009 to increase the confidence level of the detected lakes.

In my previous review, I pointed out that the spatial pattern of surface change was much more irregular than what we have seen in Antarctic subglacial lakes. In their rebuttal, the authors contend that under thin ice, the pattern of change associated with a subglacial lake can have sharp gradients, and cite two studies that looked at subglacial lakes that were under thin ice at the edges of glaciers. However, most of the lakes in this study are far from the edge of the ice sheet, and many are under ice of considerable thickness. The authors need to evaluate the ice thickness of their lakes and consider whether it makes sense that there would be large spatial variability for the locations they are considering.

Response:

    We interpolated Bedmachine v5 ice thickness at our potential lake locations (Table S2, column 'Thickness'). The ice thickness for the lake in the Hodgson paper is only about 50 m (their Figure 4), which is much thinner than the vast majority of ice beneath our lakes. While the transition may not be as smooth as for deep subglacial lakes in Antarctica (more than 2000 m), we might expect it to be smoother than that shown in Fig. R1a.

In their response to my comment about uncertainties in the Watta algorithm, the authors again seem to interpret an elevation difference between a filled and a drained lake as evidence of subglacial lake activity (figure R3). Even using the Watta algorithm, most of the profile for 5 August 2019 is on floating lake ice, and the Watta algorithm only measures the bottom of the lake in a couple of small sections where the edge of the floating ice has melted. In these places, the bottom elevation is fairly close to the profiles from 2 Aug 2020 and 1 Nov 2020. The elevation change between 2 Aug 2020 and 1 Nov 2020 is fairly substantial (+5-10 m) but this is in an area where snowfall can be heavy, and it is not implausible that a local basin in the ice-sheet surface could trap a considerable amount of snow.

Response:

We used a Figure to show how ICESat-2 can penetrate through the water column of supraglacial lakes to measure the lake bottom (Figure 2). We used example without an ice lid here (Lake NIOGHALVFJERDSFJORDEN01) in the new Figure S1.

The authors quote the Fair et al study to say that the Watta algorithm can only measure ~7 m water depth, and quote Pope et al, 2016 to say that lakes are shallow (<10 m), but their own figure S1d shows a lake that is clearly more than 15 m deep (I'm assuming they too the refractive index of water into account in interpreting the apparent depth in S1c). These assumptions don't seem to be valid and should not be relied upon.

Response:
It is right that supraglacial lakes are not necessarily shallower than 10 m. Datta et al. (2021) detected 5 lakes that are 10-15 m depth (supplementary table in their paper). In addition, Hsu et al. (2021) used the ATL03 data to derive water bathymetry for lakes (depth < 20 m). Therefore, ICESat-2 has the potential for detecting some surface lakes that are deeper than 7 m. We rephrased these statements in the revised manuscript.

One thing I don't see in the manuscript is much critical assessment of the data. An example of this is the left panel of figure 1, where the authors plot a time series of elevations from Academy_01. This time series shows a gradual gain in elevation from 2012-2019, followed by a decline after 2019. The time series, however, includes several large upward and downward spikes in elevation, that are not explained in the text or in the caption. How do the authors interpret these spikes? I would suggest that they most likely represent errors in DEMs, but I don't see that the authors recognize this, or that they acknowledge the possibility that other sharp features in time series for other lakes might be the result of DEM (or ICESat-2) errors. The authors need to acknowledge that the data that they are working from are fallible and need to explain how they differentiated between errors in the data and real signals.

Response:
We used a Hampel filter to remove the outliers in the time-series, and we acknowledge that some errors may remain particularly with the ArcticDEM. However, the long-term trend of lake activity (e.g., quiescent at high stand) can still be identified.

Throughout the manuscript, the authors present elevation changes normalized to rates of change. This does not seem appropriate for changes that are episodic (i.e. seasonal drainage and filling of lakes), and especially in table S3 it makes the data difficult to compare against each other. Unless there is a good reason to the contrary, most of the changes should be presented as elevation differences, not elevation rates.

Response:
We added a column of elevation range (the difference between maximum elevation and minimum elevation) in Table S2 to describe the magnitude of elevation change during the ICESat-2 period. We have changed the elevation-change rate to elevation differences in Table S3.

Figure 1 presents a really unusual lake as if it were a typical lake for the study. This lake is very large compared to the others in the study, and has a large, obvious, subglacially-driven change that is not typical of the other lakes, where the subglacial-vs-supraglacial difference is much less clear. It would be a much better use of space to present one or more ambiguous cases, and explain how each was interpreted, especially as the authors are claiming to make the very difficult (arguably impossible) distinction between change in supraglacial lakes and change in subglacial lakes paired with supraglacial lakes.

Response:

We have presented different scenarios of subglacial-vs-supraglacial difference in Figure 2, and have published all of the elevation profiles and relative elevation anomaly time-series in the Supplementary section (Figures S4 to S21).

I don't understand the time series presented in figure 2. There don't seem to be enough points in the right-hand column relative to the number of ICESat-2 measurements in the left-hand column. Further, the profiles in 2a seem to show the lake filling, while the time series in 2b seems to show the lake draining during the ICESat-2 period.

Response:

We have checked all data in the elevation profiles and the relative elevation anomaly. We only sampled the ArcticDEM DEMs at the locations of the ATL11 measurements to reconstruct a time series in the revised manuscript.

The authors responded to some of my questions in their rebuttal without making corresponding changes in the manuscript (see the question of which lakes were sampled by RES).

Response:

We have made the RES change. 3 of the 18 active lakes were sampled by RES data from 2017 to 2019, but no classic flat reflections were identified (Page 7, Line 210-212). Lake names are ACADEMY02, HAYES_GLETSCHER_N_NN01, and STEENSTRUP-DIETRICHSON01.

Editor

(1) Many arguments of the paper and discussion around it are related to textual descriptions of geometrical features. I thought that a cartoon explaining some basics of laser-light penetration/reflection from an area with/without floating ice, filled/drained lakes (you name it) would help. It will make all discussions and attributing different features to one or another case not only easier, but also will make it clear which characteristics are key for diagnosis in this and follow-up studies.

[Figure]

**Figure R1.** Example of how ICESat-2 can penetrate through the water column of supraglacial lakes to measure the lake bottom. The solid and dashed lines indicate strong and weak reflections, respectively. Figure (b) shows an example of a surface lake with floating ice. ICESat-2 can only penetrate the lake surface, but reflects directly off the floating ice. Figure (c) shows an example of a drained lake that ICESat-2 directly measures the ice surface. The second row (d-f) shows examples of the ICESat-2 photon reflection for the corresponding schematic.

(2) Among minor suggestions by Reviewer #1, it was written that Fig. S4 is missing. However, as I checked your supplementary material file, I did see a histogram, so it was some misunderstanding.

Response:
    We have checked the supplementary material file to make sure the elevation-range figure is included.

(3) Line 201 is shown in blue color.

Response:
    We have changed the color to black.

---

## Author Response (AR3)

Subglacial lake activity beneath the ablation zone of the Greenland Ice Sheet

Reviewer #1
The authors have cleared up a lot of the material in the study that confused subglacial and supraglacial lakes, and it seems that what remains is a much more accurate depiction of the record of possible subglacial lake activity in Greenland. I would note that most of the plots shown in the supplemental material depict basins on the surface of the ice that are getting shallower over time, which is what we would expect to see in an empty depression on the ice surface gradually being refilled by ice flow in response to the stress anomaly associated with the basin. It would be good to note that there is some uncertainty in the interpretation of lake filling rates because of this. It also would be worth pointing out that a closed depression in the ice surface in the ablation zone can be expected to fill with water during the summer, so if these depressions remain empty, it is likely that they contain moulins or crevasses that allow the water to escape, potentially to the base of the ice sheet.

Response: We thank the reviewer for the helpful feedback, we are appreciative of his or her help and time. We have added the explanation of the uncertainty source as suggested in Section 4.3.

Beyond this, my comments are mostly editorial.

Line 38: Should say that that some lakes produce a radar signature (not all)
Response: We specified that some stable lakes will produce a radar signature here.

Line 44: Gray et al, 2005 did not generate DEMs. It would be better to say that the study used radar interferometry.
Response: We corrected it to 'radar interferometry' here.

Line 75: Are these 511 RGTs the ones that intersect Greenland? Should specify what set this is.
Response: These 511 RGTs are the ones that intersect Greenland. We have specified it here.

Also line 75: I'm not sure what track segments are here—the number seems to be about double the number of RPTs. It can probably be removed.
Response: We have removed it.

Line 78: The point density does not seem like a useful metric here, because the points are not evenly distributed. I'd recommend omitting this.
Response: We have removed it.

Paragraph around line 75: I recommend splitting this into two paragraphs, one about ATL11 and another about ATL03. I think the point counts, etc relate to ATL11, and the transition to ATL03 would be cleaner with a new paragraph.

Response: We have divided it into two paragraphs.

Line 88: "was variable" should be "is variable"
    Response: Revised.

Line 109: The reference to Willis et al does not explain how the visual interpretation was done. A sentence or two for how the authors did this would be helpful.
    Response: We added the explanation here. We retained the profiles that exhibited gradual elevation change with time.

Line 133: Explain what a Hampel filter is, and provide a citation
    Response: We added the explanation and the citation here, and the revised sentence is 'A Hampel filter is a type of outlier detection filter commonly used in data analysis, and it works by identifying data points that are significantly different from their neighboring data points and replacing them with a more representative value based on the surrounding data (Hampel, 1974). Therefore, it was used to remove the outliers in the time-series of the combined ArcticDEM and ICESat-2 periods (2009-2020).'.

Line 136: I don't think "internal accuracy" is the right term here. Better to say that you assumed that the relative errors between points in each dataset were the values provided in the cited studies.
    Response: We have changed 'internal accuracy' to 'relative error' here.

Line 148: "a double reflection of both the water surface and the ice surface": this implies that there is a double reflection from each surface (four reflections in total)
    Response: We have corrected it to 'a double reflection of the water surface and ice surface beneath'.

Line 149-150: "ICESat2 can therefore be used…more than 10m deep (Fair et al, 2020)." This statement does not make sense as written, and is not supported by the citation. Should rewrite or delete.
    Response: We have removed it.

Line 156-157: Where is the result for the validation of the Watta-derived depths? It's not in the current study. This sentence needs to be rewritten or deleted.
    Response: The validation was done by Fricker et al. (2020), the Watta-derived depths are close to the LandSat-8/ Sentinel-2 -based depths and manual-picked depths from the ATL03 photons.

Line 208: "the corresponding periods": should be "the ICESat-2 period"
    Response: Revised.

214: "ICESat-2 footprints" should be "ATL11 reference points"
    Response: Revised.

247: Please explain or delete the reference to Livingstone et al (it's not clear why this is cited here. Does the current result agree with their result? Disagree?)

Response: We have removed the reference here.

259: Please elaborate on "filling activities of 60" and explain how the significance test was carried out. The null hypothesis needs to be clearly defined when significance tests are provided.

Response: We have rephrased it to 'A total of 60 lake filling events (positive volume change) that occurred from 2009 to 2020, and the volume-change rates have a positive correlation (correlation coefficient of 0.40, $p < 0.01$) with the cumulative runoff estimates.'. The significance test was provided during the calculation in the Person correlation coefficient.

Figure 4b: Does the color bar need to extend as far as 20km2? None of the lakes are that large.

Response: We have set the upper limit of the color bar to 10 $km^2$.

Figure 4d: Need to explain the pie charts

Response: We explained it as follows 'The pie chart in panel (d) represents the overall volume-change rate of lakes in each basin (blue for positive, red for negative), with the pie size is proportional to the magnitude of the absolute rate.'

4d: need to be clear about the units on the colorbar: are these dV/dt?

Response: We have changed the legend to 'dV/dt' for clarity.